# Physiological and Proteomic Responses of Cassava to Short-Term Extreme Cool and Hot Temperature

**DOI:** 10.3390/plants11172307

**Published:** 2022-09-03

**Authors:** Supranee Santanoo, Kochaphan Vongcharoen, Poramate Banterng, Nimitr Vorasoot, Sanun Jogloy, Sittiruk Roytrakul, Piyada Theerakulpisut

**Affiliations:** 1Department of Biology, Faculty of Science, Khon Kaen University, Khon Kaen 40002, Thailand; 2Department Science and Mathematics, Faculty of Science and Health Technology, Kalasin University, Kalasin 46230, Thailand; 3Department of Agronomy, Faculty of Agriculture, Khon Kaen University, Khon Kaen 40002, Thailand; 4Functional Ingredients and Food Innovation Research Group, National Center for Genetic Engineering and Biotechnology, National Science and Technology Development Agency, Pathum Thani 12120, Thailand

**Keywords:** temperature, cool stress, heat stress, cassava genotypes, photosynthesis, chlorophyll fluorescence, proteomics

## Abstract

Temperature is one of the most critical factors affecting cassava metabolism and growth. This research was conducted to investigate the effects of short-term exposure to extreme cool (15 °C) and hot (45 °C) temperature on photosynthesis, biochemical and proteomics changes in potted plants of two cassava cultivars, namely Rayong 9 and Kasetsart 50. One-month-old plants were exposed to 15, 30, and 45 °C for 60 min in a temperature chamber under light intensity of 700 μmol m^−2^ s^−1^. Compared to the optimum temperature (30 °C), exposure to 15 °C resulted in 28% reduction in stomatal conductance (gs) and 62% reduction in net photosynthesis rate (P_n_). In contrast, gs under 45 °C increased 2.61 folds, while P_n_ was reduced by 50%. The lower P_n_ but higher electron transport rate (ETR) of the cold-stressed plants indicated that a greater proportion of electrons was transported via alternative pathways to protect chloroplast from being damaged by reactive oxygen species (ROS). Moreover, malondialdehyde (MDA) contents, a marker related to the amount of ROS, were significantly higher at low temperature. Proteomics analysis revealed some interesting differentially expressed proteins (DEPs) including annexin, a multi-functional protein functioning in early events of heat stress signaling. In response to low-temperature stress, AP2/ERF domain-containing protein (a cold-related transcription factor) and glutaredoxin domain-containing protein (a component of redox signaling network under cold stress) were detected. Taken together, both cultivars were more sensitive to low than high temperature. Moreover, Rayong 9 displayed higher P_n_ under both temperature stresses, and was more efficient in controlling ROS under cold stress than Kasetsart 50.

## 1. Introduction

The effects of climate change are resulting in atmospheric warming, global temperature increase, changing wind events, erratic rainfall patterns, increased frequency of drought and prolonged dry season, and increased temperature extreme events [1,2,3]. The extent of climate change is particularly severe in the tropical regions. The interactions between plants and environments are key determinants of yield production worldwide [4,5]. Therefore, understanding how environmental changes may influence plant interactions is critical to predicting future trends of crop yield. Plant performance in terms of growth, development, photosynthesis, and yield depends on acclimation ability to the environmental changes and stresses, exercising specific tolerance mechanisms that involve a complex network of biochemical and molecular processes [6]. Abiotic stress such as extreme temperatures cause significant limitations to agriculture, more than halving average yields for major crop species [7]. The optimum temperature for cassava growth and photosynthesis varies from 25 to 35 °C [8], while the temperature below 17 °C or above 37 °C delayed sprouting of shoots and leaves of cassava [9]. In tropical savanna climate, temperature has a significant impact on the vegetative growth and development of the cassava genotypes. Phoncharoen et al. [10] reported that the lower minimum temperature (23.2 °C), and shorter day length (12.1 h) in the cool season delayed the first forking date and led to decreased total crop dry weight.

In tropical countries such as Thailand, cassava is grown at any time during the year, and the growth duration from planting to harvest is usually 12 months [11]. Therefore, plants at different growth stages are prone to being exposed to temperatures outside the optimum, causing daily short-term temperature stress during the afternoon in the hot season and early morning hours in the cool season [12]. During the 2015–2016 crop growing season in Thailand, cassava plants were exposed to the highest temperature of approximately 43 °C at 14.30 h in the hot season, and the lowest temperatures of 9 °C at night and 15 °C at 8.30 h in the cool season [13]. These extreme temperatures affected negatively physiological processes, photosynthesis, productivity, and reproduction [14]. The optimum temperature for cassava photosynthesis ranged from 25–35 °C, with net photosynthesis rate (P_n_) declining for temperatures below and above this range [15,16]. Photosynthesis performance of cassava was adaptable to its growth conditions. For example, cassava cv. M Col 1684 grown under the cool climate (mean annual temperature of 17 °C and temperature range between 12–22 °C) had maximum P_n_ at 16 μmol CO_2_ m^−2^ s^−1^ whereas those grown in a warmer climate (mean annual temperature of 24 °C and temperature range between 17–30 °C) had maximum P_n_ of 35 μmol CO_2_ m^−2^ s^−1^. Similarly, Sage et al. [17] reported that cassava cv. BGM1721 growing at 15, 30, and 35 °C under greenhouse conditions had P_n_ of 10, 24, and 39 μmol CO_2_ m^−2^ s^−1^, respectively. In an irrigated field in the tropical climate of Thailand, maximum photosynthesis rates (P_nmax_) of cassava cv. Rayong 9 were similar (27 to 30 μmol CO_2_ m^−2^ s^−1^) across the three seasons (rainy, cool, and hot season) [18]. Photosynthesis performance of cassava in temperate regions tended to be lower than in the tropics. For example, during summer in a temperate zone in Japan where the temperature conditions were comparable to the cool season in Thailand, P_nmax_ of cassava was reported at 21–25 μmol CO_2_ m^−2^ s^−1^ (before and after fertilizer application) [19]. In the rainy and cool season, in the tropics, cassava attained maximum P_n_ around noon, while in the hot season, P_n_ reached the peak as early as 9.00 h after which the stomata closed to avoid water and high-temperature stress [12]. Diurnal measurements of photosynthesis revealed the prominent inhibitory effects of pre-noon high temperatures, even for cassava growing under full irrigation. The temperature increases from 30 to 40 °C (from 8.30 to 10.30 h) in the hot season resulted in a dramatic reduction in P_n_ from 21.40 to 14.98 μmol CO_2_ m^−2^ s^−1^ despite an increase in light intensity. On the other hand, in the cool season, lower temperature of approximately 23.5 °C at 8.30 h resulted in cassava displaying P_n_ of approximately 17.17 μmol CO_2_ m^−2^ s^−1^ [20]. As revealed by chlorophyll fluorescence analysis, high-temperature stress in the hot season resulted in a reduction in effective quantum yield of PSII photochemistry (ΦPSII) and an enhancement in non-photochemical quenching (NPQ) which is one of the most important mechanisms protecting the photosynthetic apparatus, particularly photosystem II (PSII), from being damaged by reactive oxygen species (ROS) [12,20,21]. Low-temperature stress under greenhouse conditions resulted in a reduction in ΦPSII and an increase in NPQ associated with significant reduction in P_n_ of *Elymus nutans* [22] and *Vernicia* spp. [23]. Investigations on cassava photosynthesis have largely been carried out in the field conditions in which the effects of temperature were confounded by other environmental factors. The current study was designed to compare photosynthesis performance of cassava under short-term low- and high-temperature stress under a controlled environment chamber to reveal the direct effects of temperatures outside the optimum on cassava physiology. 

Proteomics analysis is a powerful tool widely used to identify differentially expressed proteins (DEPs) and improve the understanding of the mechanisms underlying abiotic stress responses in plants. For cassava, the most comprehensive analyses of leaf proteomes have been conducted to elucidate mechanisms of response to drought [24,25,26], and cold [27] stress. Proteomic analysis in relation to heat stress has been conducted in a large number of crop plants, but to the best of our knowledge heat stress-induced changes in leaf proteome has not been reported in cassava. In response to cold stress at 5 °C for 10 d, An et al. [28] identified 32 DEPs of which 10 (32%) were functioning in photosynthesis metabolism while the remaining DEPs involved in carbohydrate and energy metabolism (4 proteins), defense (2), protein synthesis (4), amino acid metabolism (1), signal transduction (1), detoxifying and antioxidants (3), protein metabolism (1), chaperones (1), DNA-binding protein (1), structure (1), and unknown functions (3). Among 558 drought-induced DEPs identified by Shan et al. [25], almost 50% were localized in chloroplast, and many up-regulated proteins were enriched in photosynthesis (PSII, PSI, cytochrome b6/f complex, electron transport, and ATP synthase). Functional enrichment analysis by Ding et al. [26] also revealed that many up-regulated proteins under drought were significantly enriched in photosynthesis (including Calvin cycle, light reaction, and photorespiration). Proteomic analysis of DEPs upon heat stress in several plants revealed that proteins relating to photosynthesis, signal transduction, antioxidant defense, transcriptional and posttranscriptional regulation, protein synthesis and turnover, carbohydrate and energy metabolism were found to play pivotal roles in protecting leaves against heat stress [29]. These reports highlighted the importance of regulation of photosynthesis and antioxidant defense during abiotic stress to ensure survival and continuing growth. 

Cassava can be grown throughout the year in the tropics, so they are unavoidably exposed to daily short-term temperature extremes which are likely to occur more frequently as a consequence of climate change. To date, there have been numerous reports investigating the effects of environmental variables related to climate change such as drought [24,30], CO_2_ enrichment [31], and nutrient deficiency [32] on cassava physiology, focusing on photosynthesis and associated parameters. Surprisingly, there has been relatively little research into the effects of short- or long-term extreme temperature stress on cassava physiology. In this study, changes in photosynthesis performance and related physiology (proline, lipid peroxidation, and membrane stability) in response to short-term low- and high-temperature stress were characterized in two high-yielding cassava cultivars. In addition, DEPs under each extreme temperature event identified in this study may help improve the understanding of the early temperature response processes, leading to the observed modulation in physiological mechanisms operating to protect cassava plants from damages induced by temperature stress. 

## 2. Results

### 2.1. Photosynthetic Performance 

Photosynthetic parameters (P_n_, gs, T_r_, C_i_, WUE, and P_n_/C_i_) at the three different temperatures are displayed in Figure 1 and Appendix A. On average for the two cultivars, net photosynthesis rates (P_n_) were highest at 30 °C (15.42 μmol CO_2_ m^−2^ s^−1^), and both extreme low (15 °C) and high temperature (45 °C) resulted in significant reductions in P_n_ down to 5.91 and 7.84 μmol CO_2_ m^−2^ s^−1^, respectively (Figure 1A). However, no significant differences in P_n_ existed among cultivars at all three temperature conditions. The average stomatal conductance (gs) for the two cultivars was moderate at 30 °C (0.09 mol H_2_O m^−2^ s^−1^) (Figure 1B). Low temperature induced stomatal closure resulting in significant reduction in gs (0.065 mol H_2_O m^−2^ s^−1^; 28% reduction). In contrast, gs under heat stress tremendously increased to 0.235 mol H_2_O m^−2^ s^−1^ (2.61 folds increase compared to that at 30 °C). Plants exposed to 45 °C had extremely high T_r_ (12.90 mmol H_2_O m^−2^ s^−1^) which were approximately 4 and 6 times higher than those under normal and low temperature, respectively (Figure 1C). High T_r_ and low P_n_ at 45 °C resulted in the plants under heat stress having the lowest water use efficiency (WUE) (Figure 1D). The highest WUE was observed in plants exposed to 30 °C (4.86 μmol CO_2_ mmol H_2_O^−1^). The WUE was 40% reduced at low temperature, but hugely reduced (10 folds) at high temperature. The internal CO_2_ concentration (C_i_) was lowest at normal temperature, and significantly increased at both temperature extremes (Figure 1E). Contrastingly, the P_n_/C_i_ ratios which reflect the efficiency of CO_2_ fixation were highest at normal temperature, and dramatically reduced at both extreme temperatures (Figure 1F). Genotypic differences in photosynthesis parameters were found in the gs and C_i_ values at 30 °C, i.e., the values of Kasetsart 50 were significantly higher than Rayong 9. On the other hand, P_n_/C_i_ at 30 °C of Rayong 9 was significantly higher than that of Kasetsart 50. 

### 2.2. Chlorophyll Fluorescence Parameters

Significant differences in chlorophyll fluorescence parameters were also observed in plants exposed to different temperatures (Figure 2 and Appendix A). The maximum quantum yield efficiency of photosystem II in the light (Fv′/Fm′) was highest at 30 °C (0.49–0.5), slightly but not significantly reduced at 45 °C (0.46–0.47), but significantly reduced at low temperature (0.43) (Figure 2A). The effective quantum yield of PSII photochemistry (ΦPSII) was highest in the plants exposed to normal temperature (0.18–0.19), followed by those at low temperature (0.12), and lowest in those at high temperature (0.5–0.6) (Figure 2B). Electron transport rates (ETR) were calculated from ΦPSII, therefore showed similar pattern of responses to temperature (Figure 2C). No significant genotypic differences in Fv′/Fm′, ΦPSII and ETR were observed at all three temperatures.

### 2.3. Electrolyte Leakage, Proline, and Malondialdehyde

The effects of temperature on electrolyte leakage (EL), proline, and malondialdehyde (MDA) of leaves of cassava cvs. Rayong 9 and Kasetsart 50 are shown in Figure 3 and Appendix A. For EL values, the significant (*p* < 0.05) difference among treatments were not observed (Figure 3A). At the low temperature (15 °C), EL of Rayong 9 (74%) was significantly (*p* < 0.05) higher than that of Kasetsart 50 (53%). Significant difference in proline content (Figure 3B) when plants were exposed to different temperatures was observed only in Rayong 9 which had highest proline at 30 °C, followed by that at 45 °C, and lowest at 15 °C. Proline contents of Rayong 9 at 30 °C (6.83 µg g^−1^) and 45 °C (5.55 µg g^−1^) were significantly higher than those of Kasetsart 50 which were 3.54 and 2.86 µg g^−1^ at 30 °C and 45 °C, respectively. In contrast, the proline content of Kasetsart 50 (3.28 µg g^−1^) at low temperature was significantly higher than Rayong 9 (1.79 µg g^−1^). For both cultivars, the MDA contents at low temperature were significantly (*p* < 0.05) higher than those at normal and high (Figure 3C). At 15 and 30 °C, Kasetsart 50 had significantly higher MDA than Kasetsart 50.

### 2.4. The Effects of Temperature on Proteomes of Cassava Leaves

As summarized in Figure 4, the number of unique proteins in Rayong 9 leaves increased from 3604 at normal temperature (30 °C) to 3636 under cool stress (15 °C). In contrast, cool stress caused a slight reduction in the number of proteins from 3726 to 3723 in Kasetsart 50. Heat stress at 45 °C caused a reduction in number of proteins in both cultivars. Under heat stress, the number of proteins reduced from 3604 to 3589 in Rayong 9, and from 3726 to 3664 in Kasetsart 50. The differentially expressed proteins between genotypes at each temperature are identified and summarized in Table 1 and Appendix A. Under cool stress at 15 °C, Rayong 9 and Kasetsart 50 expressed 2 proteins (AP2/ERF domain-containing protein and glutaredoxin domain-containing protein) which were differentially expressed and not detectable at 30 °C. The AP2/ERF domain-containing protein was the DNA-binding protein with DNA-binding transcription factor activity, while the glutaredoxin domain-containing protein showed disulfide oxidoreductase activity. The differential proteins under cool stress which were observed only in Rayong 9 were thymidine kinase, cytokinin dehydrogenase, and one uncharacterized protein (protein ID: A0A2C9VTU6). The biological functions of thymidine kinase were involved in DNA biosynthetic and thymidine metabolic process, while cytokinin dehydrogenase was involved in the cytokinin metabolic process and oxidation-reduction process. For Kasetsart 50, only one uncharacterized protein was detected. Under heat stress at 45 °C, the differentially expressed protein that was detected in both genotypes and not detectable at normal and low temperatures was annexin. The gene ontology of annexin showed many biological processes, including calcium ion transmembrane transport, phloem sucrose unloading, primary root development and response to cold, heat salt stress, and water deprivation. The differential protein, which was observed only in Kasetsart 50, was an uncharacterized protein (ID A0A2C9U0M7). However, no differential proteins in Rayong 9 under heat stress were observed.

## 3. Discussion

### 3.1. Short-Term Low- and High-Temperature Stress Induced Different Patterns of Changes in Photosynthesis and Related Physiology

Cassava photosynthesis follows a C_3_ pathway with maximum photosynthetic rate ranging from 13–24 µmol CO_2_ m^−2^ s^−1^ under chamber conditions or greenhouse [33,34]. Temperature is one of the most critical factors affecting cassava development and photosynthetic rate [10,35]. The optimum temperature for cassava growth was reported between 25–29 °C, but it can tolerate temperatures varying from 16–38 °C [36,37]. Our previous observations showed that, in the tropical field conditions in northeast Thailand, cassava plants experienced temperatures as low as 8.9 °C during the early morning hours in the cool season, and as high as 43 °C during afternoon in the hot season [13]. Cassava is a tropical root crop and sensitive to low temperature; therefore, P_n_ at 15 °C was reduced by 62% compared to that under normal temperature (30 °C) (Figure 1A). Net photosynthesis rate (P_n_) and transpiration rate (T_r_) directly related to the stomatal conductance (gs) which is the measure of stomatal opening. The stomatal pore is the first diffusional barrier for the influx of CO_2_ and efflux of H_2_O for photosynthesis and transpiration, respectively [38]. The primary functions of stomata are to maintain leaf water status during photosynthetic process by limiting transpiration rate to values that are neither wasteful nor exceeding water influx from the xylem [39]. The stomata were thus optimally opened at 30 °C, leading to both cassava genotypes displaying the best photosynthesis performance. Significantly lower P_n_ under low-temperature stress (15 °C) was primarily caused by partial stomatal closure as indicated by the 28% reduction in gs. However, P_n_ at low temperature was also limited to some extent by non-stomatal factors as indicated by significantly higher C_i_ values compared to those at 30 °C (Figure 1E). It was noted that at 30 °C, Rayong 9 had lower C_i_ (and higher P_n_/C_i_) than Kasetsart 50 (Figure 1F) indicating better efficiency of CO_2_ utilization [40]. The reduction in P_n_ and increase in C_i_ under low-temperature stress was also reported in two cultivars of *Elymus nutans* [22]. These authors found that both genotypes were similarly affected by stomatal limitation, but the less cold-tolerant genotype had much higher non-stomatal limitations together with significantly lower activity of Calvin cycle enzymes. In another study, Zhang et al. [23] reported that a dramatic reduction of P_n_ under short-term low-temperature stress in two species of *Vernicia* was caused by stomatal limitation, while both stomatal and non-stomatal factors contributed under longer exposure time. 

Contrastingly, gs of plants exposed to 45 °C were strikingly higher than those at 30 °C (2.61 folds higher) resulting in an almost four times higher transpiration rate to dissipate heat to avoid building up of temperature to the level that could cause permanent damage to the leaf. Despite the widely opened stomata, hence unlimited CO_2_ supply, P_n_ under high-temperature stress was reduced by approximately 49% compared to that at 30 °C. The reduced photosynthesis performance under high-temperature stress was therefore mainly caused by non-stomatal limitations as indicated by several folds increase in C_i_ (Figure 1E). Similar observations, i.e., an increase in gs but a reduction in P_n_ along with an increase in C_i_, were reported in *Salicornia ramosissima* after exposure to high temperature (40/28 °C, day/night, for 3 days), and reductions in P_n_ were ascribed to mesophyll and biochemical but not stomatal limitations [41]. In sorghum, the reduction in P_n_ was controlled by stomatal limitation when exposed to 40 °C (for 1 and 3 h), but when the temperature was raised to 45 °C (for 1 and 2 h) non-stomatal limitations became dominant which was in line with the current study [42].

Non-stomatal factors contributing to reduction of photosynthesis performance under stressful conditions involve a variety of processes including diffusional limitations of CO_2_ from intercellular space to the site of CO_2_ fixation in chloroplast stroma, reduced activity of Calvin cycle enzymes, retardation of electron transport, reduced rate of triose phosphate regeneration, and reduced photochemical efficiency, particularly at PSII [43,44]. In this study, both temperature extremes affected PSII photochemical efficiency as indicated by significant reductions in Fv′/Fm′ and ΦPSII (Figure 2A,B). The ΦPSII values reflect the operating quantum efficiency of photosystem II (PSII) photochemistry which directly determines the electron transport rate [45,46]. The ΦPSII of plants exposed to 45 °C were dramatically reduced (3.36 folds reduced from that at 30 °C) leading to 3.4 folds reduction in ETR (Figure 2C). A similar situation was found for field-grown cassava in the northeast Thailand. Santanoo et al. [20] measured diurnal chlorophyll fluorescence of well-irrigated cassava plants (cv. Rayong 9 and Kasetsart 50) growing in the field and recorded approximately 33% lower ΦPSII during the afternoon in the hot season (the maximum temperatures ranged from 37 to 42 °C) compared to that in the rainy season (the maximum temperatures were 31 to 36 °C). In wheat, the reduction in ΦPSII during the hot afternoon hours in summer led to the slowing down of electron transfer from PSII reaction center to the primary acceptor plastoquinone (Q_A_) and a decrease in electron transport beyond Q_A_^−^ [47]. PSII is known to be one of the most heat-sensitive components of the thylakoid membrane, and the major cause of reduction in PSII photochemical efficiency upon heat stress was due to the increase in thylakoid membrane fluidity which rendered the dissociation of the PSII light harvesting complex [48]. Moreover, the oxygen evolving complex (OEC) in PSII was known to be dissociated by heat stress, which further resulted in the inhibition of electron transport from the OEC to the PSII reaction center [49]. At 15 °C, the ΦPSII also reduced significantly compared to that at 30 °C but the reduction was much less severe to that at 45 °C (Figure 2B). At low temperature, low CO_2_ availability due to stomatal closure together with lowered Calvin cycle enzymes activity led to an imbalance between energy absorption and utilization [22]. This imbalance resulted in an over-excitation of PSII and subsequently photodamage to PSII reaction center, hence, significant reduction in ΦPSII [44,50]. It is interesting to note that, despite non-significant differences in P_n_ under low- and high-temperature treatment, the electron transport rates at 15 °C were 2.29 folds higher than the rates at 45 °C. This indicated that cassava at low temperature was under more stressful conditions than at high temperature, and alternative electron transport pathways were more accelerated to prevent over-reduction of electron transport components which could lead to generation of excess reactive oxygen species (ROS). Some reported alternative electron transport pathways which were up-regulated in chloroplast under stress included cyclic electron transport around PSI, direct electron transfer from Q_A_^−^ to oxygen via plastid terminal oxidase (PTOX), electron transfer from PSI to oxygen via Mehler reaction, and oxidation of NADPH via Flavodiiron (FLV) proteins, and malate dehydrogenase [51,52].

One of the most critical factors that damages the photosynthetic apparatus under extreme temperatures is the ROS which are predominantly generated at PSII when excess photon energy is transferred to oxygen, forming the highly reactive singlet oxygen which can then convert to other forms of ROS (superoxide anions, hydroxyl radicals, and hydrogen peroxide). Over-reduction of electron transport chains, due to the lack of the final electron acceptor NADP^+^, also leads to electron leakage from Q_A_^−^ and ferredoxin to oxygen, creating superoxide anions [53,54]. Degradation of the D1 protein at the PSII reaction center and inhibition of its repair process were reported to be the prime target of ROS attack [55]. ROS can also attack membrane lipid components of chloroplast, other organelles and plasma membrane through lipid peroxidation chain reactions. MDA is widely used as a marker of oxidative lipid injury caused by environmental stress and could be used as an important indicator of lipid peroxidation reactions that occur when plants are exposed to abiotic stress including temperature extremes. In this study, high temperature led to a small insignificant increase in MDA but low temperature induced significantly higher MDA in both cultivars (Figure 3C). This result indicated that these cassava cultivars which were being bred and selected under the tropical environments in Thailand were more sensitive to cold than hot temperature. Interestingly, for the 3-month-old cassava exposed to lower temperature (7 °C) under weak light, MDA decreased after 4 h (probably due to transient response to low-temperature shock) but increased after 9 and 24 h [56] However, for in vitro-grown cassava plantlets exposed to a gradual decrease (0.03 °C/min) in temperature from 25 to 5 °C, significant increase in MDA was not observed until 10 d after stress, and the genotype with higher cold tolerance produced lower MDA [27]. Therefore, the level of lipid peroxidation induced by low temperature varied depending on temperature, duration, plant age, and growth condition, and whether the treatment was imposed gradually or abruptly. In any case, the genotype which produced lower amount of MDA (such as Rayong 9) could be inferred to be more tolerant to low temperature. Cold stress (1 °C for 4 h) in grapevine [57] also induced significantly higher MDA contents than the controls grown at 22 °C, and the amounts of MDA were related to an increase in electrolyte leakage. However, the observed degree of lipid peroxidation in cassava in this study was not high enough to induce significantly increased membrane damage as indicated by nonsignificant differences in electrolyte leakages probably due to the short period of exposure.

Proline accumulation is widespread in response to environmental stress including extreme temperatures and generally implicated to confer stress tolerance in many plant species [58]. Proline is a multifunctional amino acid with a wide range of cellular functions including osmotic adjustment, metal chelators, ROS scavengers, redox balance, and signaling [59]. When plants were exposed to extreme temperatures for several hours/days, proline commonly accumulated and mainly displayed protective roles as compatible osmolytes and ROS scavengers [57,60]. In cassava, An et al. [56] observed a gradual increase in proline after 3 d of cold stress (5 °C), reaching the maximum after 10 d, then starting to decrease, and no relationships between proline and the level of cold tolerance among genotypes were observed. Under heat stress, patterns of changes in proline contents in a plant species varied among genotypes. As an example, short-term exposure to heat stress (42 °C for 1 h) of 5 varieties of moth bean and 32 of their mutant lines (total 37 genotypes) revealed that 28, 5, and 4 genotypes showed the increment, decrement, and no changes in proline contents, respectively [61]. Moreover, no correlations were found between proline content and heat tolerance. In this study, proline was highest at 30 °C, and its content reduced in response to both temperature extremes, especially in Rayong 9 (Figure 3B). Recently, several authors highlighted the essential roles of the proline cycle as a shuttle of redox couples (NADP^+^/NADPH, NAD^+^/NADH, and FAD/FADH_2_) crucial for energy transfer among different cellular compartments and pathways during normal growth and development [62,63]. The proline synthesis pathway in chloroplast consumed NADPH and regenerated NADP^+^, the final electron acceptor. The regeneration of NADP^+^ through proline synthesis was particularly important under stress conditions when the Calvin cycle was downregulated. Moreover, proline oxidation in mitochondria generated NADH and FADH_2_ which provided electrons for ATP synthesis needed for all cellular biosynthetic reactions. In addition, the proline cycle provides reducing powers for several key biosynthetic pathways including oxidative pentose phosphate pathway, phenylpropanoid pathway, tricarboxylic acid cycle, and purine synthesis [64]. In this study, proline tended to play important metabolic roles in Rayong 9 under optimum temperature (Figure 3B), and inhibition of proline synthesis under both low- and high-temperature shock was coupled to increased MDA (Figure 3C), and reduction in P_n_ (Figure 1A), probably through the slowing down of NADP^+^ regeneration. However, metabolic significance of proline was less obvious in Kasetsart 50 due to non-significant differences in proline contents among all three temperature conditions. It is interesting to further investigate the modulations of proline contents in different cassava genotypes under prolonged temperature treatments. 

### 3.2. Different Early Response Proteins Were Detected under Short-Term Low- and High- Temperature Stress

Under high-temperature stress, annexin was the major protein upregulated in both cassava cultivars (Table 1). Recently, annexin, a phospholipid-interacting calcium-binding protein, has emerged as one of the most important early sensors of abiotic stress signals including heat stress [65]. In response to heat stress, it has long been known that plasma membrane-localized cyclic nucleotide-gated channels (CNGCs) were activated leading to Ca^2+^ influx causing a rapid increase in cytosolic Ca^2+^ which in turn acts as a secondary messenger that initiates the signal transduction pathways, culminating in an activation of expression of heat responsive genes [66]. A recent proteomic study in Arabidopsis presented that the amount of annexin protein 1 (ANN1) was rapidly elevated within five minutes of heat treatment [67]. Moreover, functional analysis revealed that expression of heat shock protein genes and heat shock factors was inhibited in transgenic Arabidopsis mutants deficient in annexin protein, while the plants overexpressing annexin had a greater level of heat tolerance than the wildtype. Similarly, Qiao et al. [68] observed elevated expression of annexin protein upon heat stress in rice, and overexpression of annexin in transgenic rice conferred heat tolerance at various developmental stages. According to Wang et al. [67], the critical regulatory role of annexin in early heat response involved its relocation from cytoplasm to plasma membrane, then directly forming channels for Ca^2+^ influx or activating other Ca^2+^ channel proteins, causing cytosolic Ca^2+^ spiking which subsequently conveyed heat stress signals to the nucleus for activating heat shock protein genes. In addition to annexin’s role in mediating Ca^2+^ influx, it was reported to have peroxidase activity, and hence can act as an ROS scavenger [69]. The role of annexin in modulating ROS during stress was recently demonstrated in rice when down-regulation of annexin genes led to an inhibition of superoxide dismutase (SOD) and catalase (CAT) activity [70]. The ROS-scavenging activity of annexin may be responsible for the low level of ROS, as indicated by low MDA and electrolyte leakage, under heat stress in both cassava cultivars. 

In response to short-term low-temperature treatment, two proteins, namely the AP2/ERF domain-containing and glutaredoxin domain-containing protein, were up-regulated in both cassava cultivars (Table 1). In contrast, with longer period of low-temperature exposure (5 °C, for 10 days), An et al. [28] identified 35 differentially expressed proteins in leaves of in vitro grown cassava plants. Most of the identified proteins were down- regulated and involved in primary metabolism (photosynthesis, carbohydrate/energy metabolism, protein, and amino acid synthesis), defense, antioxidants, and chaperones. Only one protein (14-3-3 protein) was found to be associated with signal transduction. The two up-regulated proteins found in this study are associated with early signal perception and transduction. The APETALA2/Ethylene responsive factor (AP2/ERF) is a large transcription factor (TF) family in plants involved in controlling developmental processes and responding to multiple environmental stimuli [71]. The AP2/ERF family of TF has recently received a lot of attention as one of the most important cold-related TF families along with the previously reported TFs such as WRKY, bHLH, NAC, MYB, and C2H2 [72]. This protein family has 40–70 conserved amino acid sequences which bind specifically to specific DNA sequences on the promoters of stress-responsive genes [73]. The most well-studied member of AP2/ERF in relation to low-temperature response is C-repeat binding factors (CBFs) that are rapidly expressed and activated through Ca^2+^ or ROS signaling pathways. The activated CBFs move into nucleus and bind to specific cis-acting elements such as DREB on the promoter regions of cold-responsive genes. Recently, it was proven that overexpression of cloned CBF gene of cassava in transgenic Arabidopsis conferred cold tolerance [74]. Therefore, up-regulation of this AP2/ERF domain-containing protein in cassava within one hour of low-temperature treatment confirmed its critical role in early sensing and signaling in response to low temperature. 

Glutaredoxins (GRXs) constitute a large family of disulfide oxidoreductases that catalyze reversible reduction of disulfide bonds of target proteins by using the reducing power of glutathione (GSH), thereby post-translationally modify the activities of target proteins involved in plant development and biotic/abiotic stress response [75]. Importantly, GRXs also function in scavenging ROS and regulating redox homeostasis [76]. Recently, Dreyer and Dietz [77] reported that different isoforms of GRXs play important roles as redox transmitters in cytosol, chloroplast, and mitochondria that link the ROS signals responding to cold stress with downstream proteins/enzymes important for controlling ROS contents. Eighteen GRX genes/proteins of CC-type were identified from cassava, six of which were associated with drought responses. One of the cassava’s GRXs genes (*MeGRXC3*) was overexpressed in transgenic Arabidopsis, and conferred drought tolerance and proved to be molecularly interacting with TGA transcription factors [78]. Therefore, GRXs regulate abiotic stress responses through directly activating proteins responsible for ROS scavengers (such as antioxidant enzymes) as well as interacting with various TFs in the nucleus, to modify gene expression related to abiotic stress responses [76]. As an example, for cold stress, the transgenic tomato overexpressing Arabidopsis GRX gene (*AtGRXS17*) had an increased level of cold tolerance associated with an increased expression of genes encoding ROS-scavenging enzymes. In addition, the expressed GRXS17 protein tagged with green florescent protein was found to migrate from cytoplasm to nucleus during chilling, reflecting its role in nuclear signaling of cold stress response [79]. Further characterization and functional analysis of these two differentially expressed proteins are challenging and may lead to a deeper insight into the mechanism of cold response in cassava. 

The differentially expressed proteins between Rayong 9 and Kasetsart 50 among the treatments are noted. The total number of proteins of Rayong 9 were increased under cold temperature. The AP2/ERF domain-containing protein, thymidine kinase and cytokinin dehydrogenase were upregulated when Rayong 9 was exposed to low temperature, while only AP2/ERF domain-containing protein was detected in Kasetsart 50 leaves. Huang et al. [80] and Kjellsen et al. [81] reported that cassava is sensitive to low temperature, and it can modify its metabolism and growth to adapt to cold stress by reprogramming gene expression to increase the ability to withstand oxidative stress and synthesis of cold-induced proteins during cold acclimation. The greater number of proteins and two extra differentially proteins detected in Rayong 9 at low temperature could be related to its less severe damage as indicated by the lower MDA and slightly higher Pn. Although the molecular functions of thymidine kinase and cytokinin dehydrogenase under cold stress have not been directly elucidated, the well-known function of thymidine kinase was related to repair of damaged DNA due to ROS during stress [82], and that of cytokinin dehydrogenase was associated with modulation of cytokinin content [83]. It was well-established that cytokinin promotes plant growth and development under normal growth conditions, and also plays important roles in enhancing abiotic stress including low temperature [84]. Similar observation that Rayong 9 was more adaptable to low temperature than Kasetsart 50 was reported by Phosaengsri et al. [85]. Rayong 9 plants had longer leaf life in the cool than in the warmer season, while in Kasetsart 50, the longer leaf life was observed in the warmer season. Further studies need to be conducted to gain a further insight into the molecular mechanisms of different strategies of temperature responses of these two cultivars.

## 4. Materials and Methods

### 4.1. Plant Materials, Growing Conditions, and Temperature Treatment

Two cassava genotypes which are widely grown in Thailand including Rayong 9 and Kasetsart 50 were selected for this study. Rayong 9 has a good plant type, high photosynthetic performance, and starch content suitable for ethanol production, while Kasetsart 50 is outstanding for better growth and wide adaptability to different regions in the country [86]. These cultivars were selected for this study based on the difference in photosynthetic performance in the field conditions in a previous experiment [87]. Plants were grown for one month in the greenhouse during March to April 2016, at Department of Biology, Faculty of Science, Khon Kaen University (16.4743° N, 102.8293° E, altitude 195 m above sea level). The maximum photosynthetically active radiation (PAR) was approximately 1300 μmol m^−2^ s^−1^, maximum temperature at noon was 34 °C, and relative humidity ranged from 47–75%. Plants were grown from stem cuttings (20 cm long) in plastic pots (25 cm height × 35 cm diameter) containing 5 kg of sandy loam soils. The experimental design was a complete block design (CBD) with six replications. Soil water status was adequately kept well-watered at field capacity (FC). The fertilization was done at the rate based on soil analysis and nutrient requirements for cassava [88]. 

For temperature treatments, one-month-old plants were transferred to a temperature chamber (VRV Corp., Ltd., Bangkok, Thailand) situated at Department of Agronomy, Faculty of Agriculture, Khon Kaen University. Six plants of each genotype were treated at 15 °C, 30 °C, and 45 °C in the chamber under PAR at 700 μmol m^−2^ s^−1^ and 50–70% relative humidity. The temperature of the growth chamber was set to stabilize at the required temperature for 60 min before the plants were placed and left standing for 60 min. After 60 min, photosynthetic performance was measured. After that, leaf samples were collected and processed immediately or stored at −80 °C for physiological and proteomic determination. 

### 4.2. Photosynthesis and Chlorophyll Fluorescence

Photosynthetic parameters including net photosynthesis (P_n_), stomatal conductance (gs), internal CO_2_ concentration (C_i_), ratio between P_n_ and C_i_ (P_n_/C_i_), transpiration rate (T_r_), water use efficiency (WUE), effective quantum yield of PSII photochemistry (ΦPSII), and electron transport rate (ETR) were measured on the second fully expanded leaf of six plants by using an Infrared Gas Analyzer (IRGA) model Li-cor 6400xt with 6400-40 leaf chamber fluorometer (Li-Cor Inc., Lincoln, NE, USA). The conditions of measurement were fixed as follows: PAR at 1500 μmol m^−2^ s^−1^, CO_2_ concentration 400 μmol mol^−1^, and VPD between 0.9 to 1.5 kPa. 

### 4.3. Electrolyte Leakage (EL), Proline, and Malondialdehyde (MDA)

The degrees of cell membrane injury induced by temperature stress were evaluated by determining the electrolyte leakage according to the method of Bajji et al. [89]. Five leaf segments (ca. 1 cm long) were put into 10 mL of deionized water in a test tube at room temperature for 24 h in the dark. Electrical conductivity (EC) of the water was then measured to obtain EC_1_ value by using an EC meter (PL-700PCS GONDO, Taiwan, China). The tube containing leaf segments were then boiled for 15 min. After the tubes cooled to room temperature, EC_2_ were measured. Electrolyte leakage (%) was calculated from the equation: EL = (EC_1_/EC_2_) × 100.

Free proline content in cassava leaves was determined following the method of Bates et al. [90]. The content of proline was extracted from 0.5 g of leaf samples in 5 mL of 3% (*w*/*v*) aqueous sulphosalycylic acid for 3 h. Two mL solvent was reacted with 2 mL of acid-ninhydrin and 2 mL of glacial acetic acid and boiled in a water bath at 100 °C for 1 h. The reaction was stopped by putting the tubes on ice. The solution was extracted with 4 mL of toluene and the absorbance of the toluene fraction was measured at 520 nm using a spectrophotometer (Hanon, Model i3, Jinan, China). The amount of free proline was evaluated using a standard curve and expressed as μg g^−1^ tissue fresh weight.

Determination of malondialdehyde (MDA) as an indicator of lipid peroxidation in cassava leaf was carried out following the method of Velikova et al. [91]. Leaf sample 0.1 g fresh weight was homogenized in 2.5 mL 0.1% (*w*/*v*) trichloroacetic acid (TCA) solution. The homogenate solution (1 mL) was centrifuged at 10,000× *g* for 20 min. Supernatant 0.5 mL were added to 5 mL of 0.5% (*w*/*v*) thiobarbituric acid (TBA) in 20% TCA. The mixture was incubated in boiling water for 10 min, and the reaction stopped by placing the reaction tubes in an ice bath for 5 min. Then, the solution was centrifuged at 10,000× *g* for 5 min. The absorbancy of the supernatant was read at 532 nm and 600 nm. The MDA content was calculated using the extinction coefficient of the red MDA-TBA product of 155 mM^−1^ cm^−1^. The MDA concentration in mM was calculated from the formula: MDA = (A532 − A600)/155.

### 4.4. Proteomics Analysis

Fresh cassava leaf samples were collected from six plants of each genotype and treatment. Briefly, 800 mg of leaf tissue from six plants was ground to a fine powder in liquid nitrogen. Total protein from 100 mg leaf samples were extracted using 0.5% sodium dodecyl sulphate (SDS). The concentration of proteins was measured according to Lowry et al. [92] using BSA (bovine serum albumin) as a standard protein and absorbance at 595 nm was recorded by an LED microplate reader (RT-6900, Rayto, Shenzhen, China). In-solution digestion and LC–MS/MS were carried out using the in-house method of the Functional Proteomics Technology Laboratory, National Center for Genetic Engineering and Biotechnology, Thailand [93]. Prior to LC-MS/MS analysis, the digested samples must be dried and protonated with 0.1% formic acid before injection into LC-MS/MS. The protocol for analysis of LC-MS/MS was described by Pan-utai et al. [94]. The trypsin digested peptides from each treatment were prepared for further analysis with LC-MS/MS (U-300 Nano-LC system and HCTultra, LC-ESI-Ion Tap MS, Thermo Fisher Scientific, Cambridge, UK). 

The quantitation of protein based on the peptide ion signal intensities of acquired LC-MS raw data was performed using DeCyder MS Differential Analysis software (DeCyder MS version 2.0, GE Healthcare, Uppsala, Sweden) [95,96]. The Mascot search engine was used to correlate MS/MS spectra to the Uniprot Manihot esculenta database. The level of proteins in each sample was expressed as log2 value. Venn diagrams were used to show the differences between protein lists originating from different differential analyses [97]. The protein spectral data used in this study have been deposited at ProteomeXchange: PXD036212 and JPST001820, (https://repository.jpostdb.org/preview/38745956663037f551dfa4, Access key 8921 (accessed on 22 August 2022)).

### 4.5. Data and Statistical Analysis

All physiological data among temperature treatments were analyzed using one-way analysis of variance (ANOVA) and the mean comparisons between treatments were done using Tukey’s test under Sigmaplot version 11.0 (Systat software, Inc., San Jose, CA, USA). For mean comparisons between plant cultivars in the same temperature treatment, Student’s *t*-test was used. Statistical significance was taken at *p* < 0.05 and *p* < 0.01. All statistical analyses followed the procedure described by Gomez and Gomez [98]. Protein identified with LC-MS was used for analysis with multiple array viewer program (MeV v4.8.1, SourceForge, San Diego, CA, USA) [99]. 

## 5. Conclusions

In comparison to leaf performance of both cassava genotypes at the optimum temperature (30 °C), short-term low (15 °C) temperature stress resulted in stomatal closure, reduced P_n_, lower T_r_, and higher C_i_. In contrast, short-term high (45 °C) temperature stress strongly induced stomatal opening to dissipate heat via dramatically enhanced transpiration while P_n_ was reduced, suggesting the dominant roles of non-stomatal limitations on photosynthesis. Cassava was more sensitive to low than high-temperature stress as revealed by lower P_n_, more intense oxidative stress (indicated by higher MDA content), and membrane leakage. Despite the lower P_n_, plants under low temperature displayed more than double ETR compared to that at high temperature, suggesting higher activity of alternative electron transport pathways to prevent excess ROS. Rayong 9 was more efficient in controlling ROS as indicated by significantly lower MDA under cold stress. Proteomic analysis detected upregulation of proteins involved in early temperature signaling (annexin under heat stress), cold-related transcription factor (AP2/ERF domain-containing protein under cold stress), and ROS signaling and scavenging (glutaredoxin domain-containing protein under cold stress). Differentially responsive physiological parameters and proteins reported in this study may be employed as potential markers for future research leading to better elucidation of molecular mechanisms underlying the response and acclimation to extreme temperature stress.

## Figures and Tables

**Figure 1 plants-11-02307-f001:**
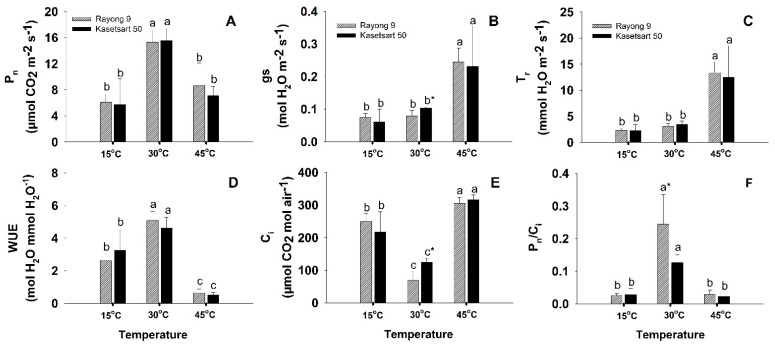
The effects of short-term exposure to different temperatures (15, 30, and 45 °C) on photosynthetic parameters including net photosynthesis (P_n_) (**A**), stomatal conductance (gs) (**B**), transpiration rate (T_r_) (**C**), water use efficiency (WUE) (**D**), intercellular CO_2_ concentration (C_i_) (**E**), and ratio between P_n_ and C_i_, and (**F**) of cassava cvs. Rayong 9 and Kasetsart 50. Means (±SD) which are significantly different (*p* < 0.05) among treatments for each cultivar are denoted by different lower-case letters. The significant differences between cassava genotypes are denoted with *.

**Figure 2 plants-11-02307-f002:**
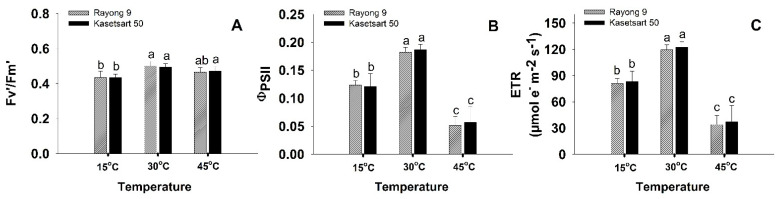
The effects of short-term exposure to different temperatures (15, 30, and 45 °C) on chlorophyll fluorescence parameters including maximum quantum yield efficiency of photosystem II (Fv′/Fm′) (**A**), effective quantum yield of PSII photochemistry (ΦPSII) (**B**), and electron transport rate (ETR) (**C**) of cassava cvs. Rayong 9 and Kasetsart 50. Means (±SD) which are significantly different (*p* < 0.05) among treatments for each cultivar are denoted by different lower-case letters.

**Figure 3 plants-11-02307-f003:**
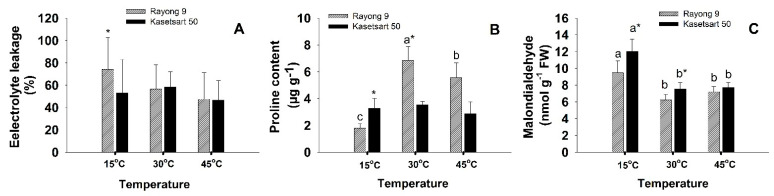
The effects of short-term exposure to different temperatures (15, 30 and 45 °C) on electrolyte leakage (EL) (**A**), proline (**B**), and malondialdehyde (MDA) (**C**) of cassava cvs. Rayong 9 and Kasetsart 50. Means (±SD) which are significantly different (*p* < 0.05) among treatment for each parameter are denoted by different lower-case letters. The significant differences between cassava genotypes are denoted with *.

**Figure 4 plants-11-02307-f004:**
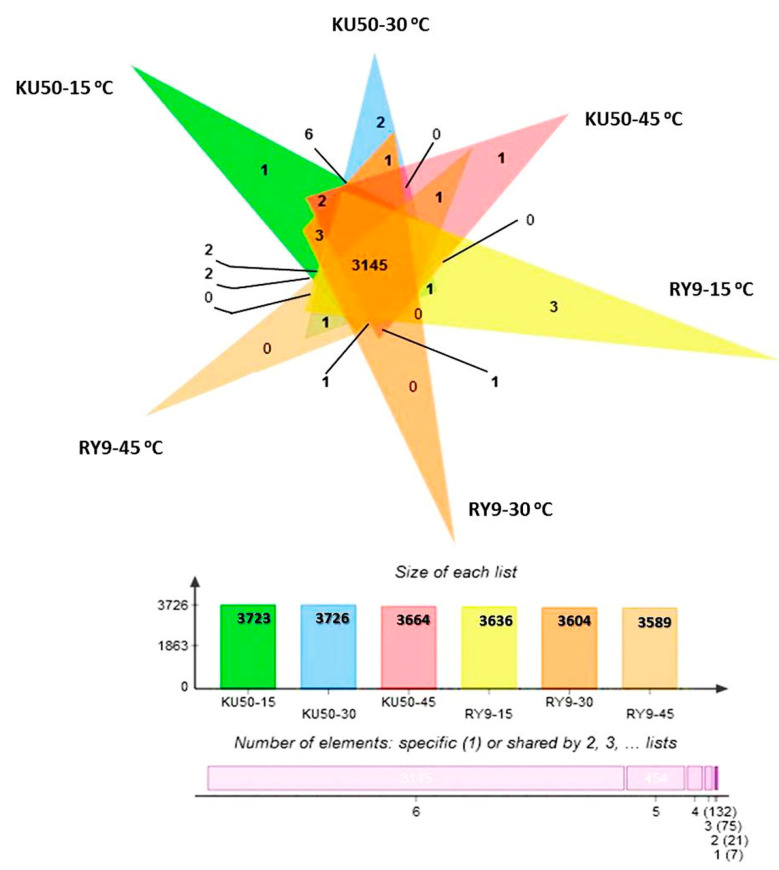
Venn diagram of proteins differentially expressed in leaves of cassava cvs. Kasetsart 50 (KU50) and Rayong 9 (RY9) after exposure to 15, 30, and 45 °C for 60 min in a temperature chamber. The number of expressed proteins in Kasetsart 50 after incubating at 15, 30, and 45 °C for 60 min was 3723, 3726, and 3664, respectively, while 3636, 3604, and 3589 proteins in Rayong 9 leaves were detected after incubating at 15, 30, and 45 °C for 60 min. The protein lists from leaves of Kasetsart 50 and Rayong 9 were input to a Venn diagram and 1 protein was identified as being from KU50 at 15 °C, 2 proteins from KU50 at 30 °C, 1 protein from KU50 at 45 °C, 3 proteins from RY9 at 15 °C, and 3145 proteins from both KU50 and RY9 at 15, 30, and 45 °C.

**Table 1 plants-11-02307-t001:** The differentially expressed proteins in leaves of cassava cvs. Rayong 9 and Kasetsart 50 exposed to low- (15 °C) and high-temperature (45 °C) stress compared to those at normal temperature (30 °C).

Genotype	Protein ID	Protein Names	Gene Ontology (Biological Process)	Gene Ontology (GO)	Cellular Component
KU50-15 °C|RY9-15 °C	A0A2C9V0E9	AP2/ERF domain-containing protein		nucleus; DNA binding; DNA-binding transcription factor activity	nucleus
	A0A2C9UPP7	Glutaredoxin domain-containing protein		cytoplasm; protein disulfide oxidoreductase activity	cytoplasm
RY9-15 °C	A0A2C9WDC0	Thymidine kinase	DNA biosynthetic process; thymidine metabolic process	ATP binding; thymidine kinase activity; DNA biosynthetic process; thymidine metabolic process	
	A0A2C9VTU6	Uncharacterized protein		integral component of membrane	integral component of membrane
	A0A2C9UA68	Cytokinin dehydrogenase	cytokinin metabolic process; oxidation-reduction process	cytokinin dehydrogenase activity; FAD binding; oxidoreductase activity; cytokinin metabolic process; oxidation-reduction process	
KU50-15 °C	A0A2C9VH14	Uncharacterized protein			
KU50-45 °C|RY9-45 °C	A0A0C5A1R1	Annexin	calcium ion transmembrane transport; phloem sucrose unloading; primary root development; response to abscisic acid; response to cadmium ion; response to cold; response to heat; response to salt stress; response to water deprivation	apoplast; cell wall; chloroplast stroma; cytosol; mitochondrion; plasma membrane; plasmodesma; thylakoid; vacuolar membrane; ATP binding; calcium ion binding; calcium-dependent phospholipid binding; peroxidase activity; protein homodimerization activity; zinc ion binding; calcium ion transmembrane transport; phloem sucrose unloading; primary root development; response to abscisic acid; response to cadmium ion; response to cold; response to heat; response to salt stress; response to water deprivation	apoplast; cell wall; chloroplast stroma; cytosol; mitochondrion; plasma membrane; plasmodesma; thylakoid; vacuolar membrane
KU50-45 °C	A0A2C9U0M7	Uncharacterized protein			
RY9-45 °C	—	—	—	—	—

Information of each differentially expressed protein was retrieved from the Uniprot database (https://www.uniprot.org/ (accessed on 18 March 2021).

## Data Availability

Data is contained within the article.

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
