# Peer review of "Physiological and Proteomic Responses of Cassava to Short-Term Extreme Cool and Hot Temperature"

_plants, 2022, doi:10.3390/plants11172307_

Round 1
Reviewer 1 Report
The manuscript „Physiological and proteomic responses of cassava to short-term extreme cool and hot temperature” by Santanoo et al. concerns the topic of short-term low and high temperature stress effect on photosynthesis, biochemical and proteomics changes of cassava plants grown in controlled environment chamber. The authors raised an important topic that has not been sufficiently studied so far. It is particularly important in terms of the observed climate changes.
The Introduction part of the article is clearly written and it includes all important information.
The experiments were conducted with valid methodologies. The ‘Materials and Methods’ section is clearly written and all measurements are well described. Only in case of subsection ‘Data and statistical analysis’ I suggest to add information about the program that was used for statistical analysis.
The results are clearly reported in the text. A valid discussion section in perspective to the results and literature has been presented. The conclusions are clearly formulated.
Although I generally find the article acceptable for the publication, there are a few things that should be improved, namely:
- In my opinion instead of ‘cool temperature’ used widely in the text ‘low or cold temperature’ should be used.
- Line 195 – ‘were showed’ or ‘were shown’?
- Supplementary Table 2 mentioned at line 221 is missing.
- Figure 4 – caption of the figure should be extended. There are illegible numbers on the block located horizontally at the bottom of the figure. It should be clearly explained what is presented on each part of the figure. Currently, the meaning of all numbers is difficult to decipher.
- Table 1 – the database/reference used to describe the proteins should be given in the table caption.
- Line 323 – missing word between ‘double’ and ‘those’.
- Line 515 – repetition of word ‘chamber’.
- Line 533 – probably word ‘injury’ should be written instead of word ‘jury’.
- Line 581 – 588 – information about the program used for statistical analysis should be added.
- Line 610 – 618 – Supplementary material is not available.
Author Response
Please see the attachment for response to Reviewer 1.

Reviewer 2 Report
The current manuscript examines the effects of short-term hot and cold treatment on two cassava cultivars, Rayong 9 and Kasetsart 50. In this study, plants were exposed to different temperatures and their stomatal conductance and photosynthetic rate were reduced. The authors also did comparative proteomics and found that some proteins were differentially expressed and correlated with stress treatment. Lastly, the authors concluded that both cultivars are more sensitive to low temperatures than high temperatures. In addition, Rayong 9 displayed a higher photosynthetic rate than Kasetsart 50. The authors made a great effort to do this work, but the manuscript still lacks very useful information.
1. Raw proteomics data is missing.
2. Data from proteomics must be submitted in PRIDE, and the user name and password for accessing the data must be included in the manuscript.
3. The conclusion made that Rayong 9 displayed a higher photosynthetic rate than Kasetsart 50. If we look at the graph showing this data, it appears to be insignificant.
4. As a result of heat and cold stress, did the author notice any changes in the seedling's growth?
Author Response
Please see the attachment for response to Reviewer 2.

Round 2
Reviewer 2 Report
I think with the revised version this manuscript can be accepted for the publication.